# Molecular Engineering of Peptide–Drug Conjugates for Therapeutics

**DOI:** 10.3390/pharmaceutics14010212

**Published:** 2022-01-17

**Authors:** Yu Fang, Huaimin Wang

**Affiliations:** Key Laboratory of Precise Synthesis of Functional Molecules of Zhejiang Province, School of Science, Westlake University, 18 Shilongshan Road, Hangzhou 310024, China; fangyu@westlake.edu.cn

**Keywords:** peptide–drug conjugate (PDC), self-assembly, nanotechnology, hydrogel, prodrug

## Abstract

In recent years, hundreds of novel small molecular drugs used for different treatments have been studied in the three phases of clinical trials around the world. However, less than 10% of them are eventually used due to diverse problems. Even some traditional drugs that have been approved by the Food and Drug Administration (FDA) have faced similar dilemmas. For instance, many drugs have poor water solubility, are easily hydrolyzed, or possess undesirable toxicity, while a variety of cancer cells develop drug resistance (DR) or multiple drug resistance (MDR) towards chemotherapeutic agents after long-term therapy. In order to improve the efficacy and efficiency of drugs, research has been directed forward towards the creation of assemblies of peptide–drug conjugates (PDCs) which have proven to possess wide potential for overcoming such complications based on their excellent biocompatibility, controllable biodegradability, site-selective targeting, and comparably low cytotoxicity. In this review, we focus on the recent developments and advances made in the creation of self-assembled nanostructures of PDCs for cancer therapy, on the chemical and physical properties of such drugs and peptides, and how they are arranged together to form diverse supramolecular nanostructures. Additionally, we cover certain mechanisms regarding how peptides or their derivatives enhance the efficiency and efficacy of those selected drugs and provide a brief discussion regarding the perspectives and remaining challenges in this intriguing field.

## 1. Introduction

Improving the efficacy and efficiency of both approved and unapproved drugs is always imperative for the delivery of treatments. Additionally, such improvements have considerable commercial value, which has drawn attention to different areas of study. Nowadays, many research groups are endeavoring to use diverse strategies to overcome the different existing problems [1,2,3]. 

Approximately 10 million deaths were caused by cancers last year and they are one of the leading causes of death all over the world [4,5]. The formation of a tumor occurs due to the uncontrollable growth of certain cells in the body [6], and according to whether they invade nearby tissues and spread to other parts in the body or not, tumors can be categorized into two types, namely malignant tumors and benign tumors [7]. Malignant tumors are cancerous and difficult to treat, since even though they can be removed from the body via surgery, it is hard to eliminate them totally and they relapse easily [8]. For cancer treatments, many approved chemotherapeutic agents and immunotherapeutic agents are toxic to healthy tissues and even to organs, because of their low site-specific targeting ability. There is a myriad of novel small molecules that have been proven to possess good anti-cancer efficiency, although these have not yet been approved by the FDA because of their complicated adverse effects or their unclear working mechanisms. Hence, many methods have been used to modify these compounds so as to improve their performance for utilization for anti-cancer therapy [9,10,11,12]. Those drugs that need to be modified may have several problems or just a key deficit that needs to be fixed, such as poor solubility in physiological environments [13], undesired toxicity [14], high dose usage [15], or poor specific targeting ability [16]. Additionally, after long-term treatments, many diseases may develop different degrees of DR or even MDR [17], including cancers [18]. Through drug modification, slight structural variations in the drugs could induce big differences in their pharmacodynamics, pharmacokinetics, and toxicity [19,20]. Therefore, drug modification is complicated and unique in each case. Nowadays, many approaches are focusing on the development of drug delivery systems. Many reported cases have combined drugs (cargo) with different materials (vehicle) via non-covalent interactions to form supramolecular nanostructures to then be delivered to their target sites. For example, drugs can be mixed with polymers to form nanoparticles, which helps them to cross the cell membranes [21]. In contrast, some other drugs are covalently conjugated with designed molecules through a cleavable linker to form prodrugs. Some of those prodrugs can self-assemble to form supramolecular complexes to prolong the drug’s release time in the body or to perform other functions. Normally, the materials used to covalently conjugate drugs are bio-inspired, such as carbohydrates [22], peptides [23], and nucleotides [24], although there are also some artificial polymers being used. Before these studies, many antibody–drug conjugates (ADCs) were studied, with some inspiring results being obtained [25]. In comparison with other materials, prodrugs constituted by peptides possess a plethora of advantages. There are more than 20 types of amino acids (AAs) that can act as the basic units for building peptides. Hence, peptides have enormous potential to be designed for different purposes. Most materials built from peptides are biocompatible and biodegradable, with low or without any unexpected toxicity [26,27]. As shown in Figure 1, peptides can chemically react with drugs to produce peptide–drug conjugates (PDCs). Most PDCs can be stimulated to self-assemble to different conformations in response to diverse stimuli or triggering forces, including stimulation by temperature [28], pH [29], enzymes [26], ultrasound [30], or ions [31], among others. Additionally, they can even self-assemble without any stimulus. PDCs normally self-assemble into different nanostructures, such as nanotubes [32], nanoparticles [33], and nanofibers [34]. Moreover, comparing peptides related to drugs non-covalently with PDCs, the self-assembly of PDCs has many advantages, such as selective targeting, long blood circulation time, controlled release, and a precise loading capacity. 

This review mainly focuses on the development of molecular engineering and the self-assembly of PDCs in recent years. Especially, it is centered on the use of PDCs as building blocks to self-assemble to form different nanostructures for diverse anti-cancer therapies and treatments of some other diseases in the last five years. Moreover, the future perspectives and limitations of the studies are also discussed with input from our own understanding.

## 2. Camptothecins

Camptothecins are a category of molecules that consists of camptothecin (CPT) and its analogues. CPT is a plant alkaloid and a well-known chemotherapeutic agent that is extracted from the bark and stem wood of the native Chinese tree *Campatotheca acuminata* [35]. This herb (*Campatotheca acuminata*) is widely used in traditional Chinese medicine. CPT was co-discovered by Professor Monroe Eliot Wal and Professor Mansukh C. Wani in the mid-20th century, when they tried to screen certain natural compounds for anti-cancer treatments [35]. Initially, a crude product from *Campatotheca acuminata* was found to be very active in the L1210 mouse leukemia life-prolongation assay [35], although the molecular structure of the active ingredient was not validated until 1966 [36]. After its structure was validated, CPT was proven to possess excellent anti-tumor efficiency in animal and preclinical trials, especially for breast, lung, and stomach cancers [37]. However, many side effects were also observed in those assays. For instance, CPT is a water-insoluble (0.551 mg/mL) molecule and has no specific targeting ability towards tumor cells, which causes damage to healthy tissues near the tumor [38]. Additionally, CPT has low tumor penetration ability, which means that only the surface of the tumor tissue can be treated. With an unclear mechanism of treatment and many other complications, eventually the clinical trials for CPT were stopped in the 1970s in phase II [35]. Nonetheless, the study of CPT has never stopped. Afterwards, the mechanisms of action for CPT were revealed. CPT can selectively combine with the nuclear DNA topoisomerase types I and DNA complexes and then stabilize them to induce the failure re-ligation of DNA, which causes DNA eradication and results in cell apoptosis [38]. Later, many artificial CPT derivatives were designed and synthesized. There are four CPT analogues that have been approved as chemotherapeutic drugs for clinical use to date, namely irinotecan [39], topotecan [40], belotecan [41], and trastuzumab deruxtecan [42]. For the formation of trastuzumab deruxtecan, a CPT derivative was connected to the antibody trazstuzumab via a peptide linker. Many other analogues have shown a number of problems, resulting in their failure at animal or clinical trial phase. In this section, we will outline the research conducted over the last five years regarding the self-assembly of conjugates of peptides and CPT.

### 2.1. Different Stratagies for Camptothecin (CPT) Modification

CPT has two formations, CPT and CPT carboxylate. CPT can be hydrolyzed to CPT carboxylate in physiological environments. CPT carboxylate has much better water solubility than CPT, although CPT seems to be the active form used for anti-cancer therapy [43]. Figure 1 illustrates CPT and its derivatives that are mentioned in this part.

In recent years, Cui and his co-workers have focused on the modification of CPT through peptides and have made some thrilling advances. The Cui group published their first work related to CPT in 2014 [44]. In that research project, the conjugation of CPT and a Tau-derived peptide (VIQVIC) [45] formed a PDC via a disulfanyl butyrate (BuSS) linker (CPT-buSS-Tau). CPT-buSS-Tau can self-assemble in water to form nanotubes, which prolongs the release time of CPT in situ and in the presence of glutathione (GSH) [46]. After Cui et al., first proposed the CPT–peptide conjugation to self-assemble to form nanostructures, the same group further developed the diCPT strategy to introduce CPT in the self-assembly system to overcome the drawbacks of CPT applications [32]. The reducible linker disulfanylethyl carbonate (etcSS) is conjugated with CPT and then modified with the targeting peptide iRGD at both the side chain of lysine and the N-terminal of the peptide to form the amphiphilic prodrug diCPT-etcSS-iRGD (diCPT-iRGD), which is also a hydrogelator (Figure 2B). The in vitro cytotoxicity assay showed that the IC_50_ of diCPT-iRGD towards GL-261 brain cancer cells is only half that of CPT, which is 229.2 nM [32]. Furthermore, diCPT-iRGD self-assembles to form nanofilaments with a hollow morphology, as evidenced by transmission electron microscopy (TEM) results [32]. In the formed nanotubes (NTs), the hydrophobic diCPT head of the compound is located on the inner surface of the NT and assists the hydrogelators to assemble through the hydrogen bond (H-bond), π-π staking, and many other non-covalent interactions. The hydrophilic cyclic peptide iRGD is located on the outer surface of the NT, and not only improves the solubility of the hydrogelator, but also has the ability to target the tumor via binding neuropilin-1 and enhancing CPT’s tumor penetration ability [47]. Since iRGD is on the outer surface of the NT, the electrostatic charge of the outer surface of the NT is positive, meaning it can be combined with the negatively charged stimulator of interferon genes (STING) agonist cyclic diadenosine monophosphate (CDA) through an electrostatic interaction, as demonstrated by the zeta potential analysis [32]. Furthermore, the CDA-loaded NT (CDA-NT) can form self-supporting hydrogels by increasing its concentration or adding counterions (supplied by PBS or cell culture medium) [32]. The in situ experimental results showed that after mixing CDA with the diCPT-iRGD NT aqueous solution and subcutaneously injecting the mixture into the mice, a spherical-shaped hydrogel formed around the tumor tissues. Since the concentration of GSH is much higher in intratumoral environments than in other tissues, the CPT can be released from the hydrogel via the cleavage of the ectSS linker by GSH to induce the degradation of the hydrogel, followed by a release of CDA. The release process has been proven to last for about 20 days in situ, which is a much longer time than in the administration of CPT and CDA [32]. With the release of CDA, a variety of immune responses are activated to hamper tumor activity. As previously mentioned, CPT can interact with the nuclear DNA through diverse mechanisms to inhibit the tumor, a process that may also release fragments of DNA into the cytosol to stimulate the intrinsic STING-dependent pathway [32]. Additionally, in comparison with systemic chemotherapy involving CPT, the use of a hydrogel to release CPT locally may reduce the systemic-chemotherapy-associated immunosuppression and promote the activation of dendritic cells and tumor infiltration of T cells to clear the tumor [32]. It has also been demonstrated that STING signaling pathways can enhance the tumor’s immunogenicity and can be stimulated via binding with cyclic guanosine monophosphate and adenosine monophosphate (cGAMP) [32]. After CDA is released from the hydrogel system, it can be utilized to synthesize cGAMP, activating the STING signaling pathway to induce the production of type I interferons and other cytokinases. Eventually, the activation of the STING pathway in the tumor can also activate the tumor-antigen-specific T cells, leading to natural killer cells infiltrating in the tumor and mobilizing an anti-tumor response [32]. Although CDA by itself has high toxicity for cells, it does not have the ability to selectively target tumor cells and has high degradability in physiological environments [48], meaning its use with diCPT-iRGD for in situ gelation is a brilliant anti-tumor therapy. From the results of animal trials, tumors in the volume range of 100 to 150 mm^3^ could be totally cleared by the CDA-NT hydrogel in approximately 15 days [32]. According to both the cell experiments and animal trials, the CDA-NT hydrogel demonstrated an excellent anti-tumor ability with low cytotoxicity to healthy tissues. In contrast with the administration of CDA alone in situ, which can be totally cleared in 5 days, the CDA in the CDA-NT can last longer than 20 days in situ. Additionally, the Cui group also employed a matric metallopeptidase 2 (MMP2) responsible peptide PLGLAG between diCPT and iRGD to form diCPT-PLGLAG-iRGD, which can also form NTs [49]. In this case, the anti-PD1-antibody (aPD1) was selected to be mixed with diCPT-PLGLAG-iRGD [49]. As with the diCPT and CDA mixture, this mixture can also immediately form a hydrogel (P-NT) in PBS or cell culture medium. With the enzymatic and GSH induction, aPD1 and CPT can be slowly released for anti-tumor therapy with neglectable adverse effects (Figure 2A). Furthermore, both CDA-NT and P-NT can induce a memory response for tumor inhibition, as demonstrated by the high expression levels of central memory T cells and effector memory T cells in tumor rechallenge mice models [32,49]. Above all, the peptide-modified CPT system has much better treatment performance than CPT itself, and it can also serve as the carrier for other chemo- or immunotherapeutical agents, such as small molecules, agonists, and antigens, in order to provide dual pathways for cancer therapy [32,49]. In addition, immunotherapy has become a very popular research direction in recent years and has become seen as the future of cancer treatments, especially after the first chimeric antigen receptor T-cell (CAR-T) therapy was approved by the FDA. However, the approved CAR-T therapy can only be used to treat liquid tumors. Cui et al., developed a method that combines PDCs with STING agonist and aPD1 to activate the immune response and to generate synergistic effects to clear solid tumors, which is a very creative strategy that has made much progress for cancer therapy. Additionally, the information regarding CPT that was used in these cases is well established, which could reduce the cost of further studies. 

Wang and his colleagues from Macau University designed a CPT–peptide conjugate that can be bound with cucurbit [7] uril (CB7) via a host–guest interaction [33]. In their design, CPT was covalently conjugated with a pentapeptide NH_2_-FFVLK-COOH (FFVLK) via a succinate acid (SA) linker on the side chain of lysine (K) to form a FFVLK-CPT conjugate (PC). The pentapeptide FFVLK is the reverse version of KLVFF and can self-assemble to form nanofibers through a H-bond and π-π stacking [50]. CB7 can bind with the N-terminal phenylalanine (F) on the PC through a host–guest reaction at an optimized ratio to form CB7-PC (CPC) [33]. Then, based on the hydrophilic domain CB7 and hydrophobic domain PC in this building block, CPC can self-assemble to form nanoparticles that can stably circulate in vivo [33]. Spermine is overexpressed in cancer cells, which can dramatically reduce host–guest interactions between CB7 and PC to induce the morphology transformation from nanoparticles to microfibers, then causing cytotoxicity for cancer cells [51]. According to their experimental results, the IC_50_ of CPC against 4T1 cells (breast cancer cell line) is much lower than that of CPT. However, the IC_50_ values of CPC and CPT are reversible against AML12 cells (alpha mouse liver 12 cell line) [33]. These results show that CPC has excellent bioavailability and has good specificity for cancer cell targeting. Here, utilizing spermine to induce the conformation transfer from nanoparticles to nanofibers is a brilliant idea, which means the drug is retained in the target cells. This strategy not only solves the issue of the poor solubility of CPT in physiological environments and decreases the side effects of CPT, but it also prolongs the half-life of CPT in situ.

Moreover, Wang and co-workers also made important contributions regarding CPT modifications. They modified CPT by conjugating it with different peptides, then the resulting nanostructures of PDCs were used to treat cancers through diverse strategies, achieving some success [52,53]. Additionally, a non-covalent strategy was used by them to delivery belotecan for anti-cancer therapy and the results were very promising [54]. 

### 2.2. Conjugates of Peptides with 10-Hydroxycamptothecin (HCPT) as the Prodrug for Cancer Therapy

HCPT belongs to a class of organic compounds known as camptothecins. As an analogue of CPT, HCPT has better anti-cancer ability and less side effects than CPT, although it still has poor water solubility and cancer cell targeting issues [55]. Hence, HCPT is still unapproved by the FDA and is currently in the clinical trial phase. Intratumoral injection is the most common administration method for HCPT, which may cause patient compliance issues. Figure 2 summarized HCPT and its derivatives that are mentioned in this part. Yang et al., seem to have found a way to solve the obstacles of HCPT by conjugating it with a well-designed peptide and delivering it not only into tumor cells, but also into the cell nucleus via intravenous injection [34]. In comparison with intratumoral injections, intravenous injections are much easier to operate. For intravenous injections, drugs are injected into veins then circulate in the body through the bloodstream to bind to their target sites or be cleared. On the contrary, for intratumoral injections, the volume of the tumor needs to be big enough to be detected, which is a drawback for cancer therapy at early stages, and when injecting the drug care should be taken to avoid the needle breaking any veins. In Yang’s project, HCPT was modified by covalently conjugating it with the peptide FFERGD via a glutaric acid linker at the N-terminal to form the prodrug HCPT-FFERGD (HP). HP possesses excellent solubility and forms a stable and clear solution in water (1 mg/mL) [34]. However, with a slight change in the peptide sequence to HCPT-FFARGD or HCPT-FFERGA, both form aggregates in water at the same concentration [34]. This result shows the importance of molecular engineering to balance the hydrophobicity and hydrophilicity in self-assembly systems. The platinum (II)-based alkylating chemotherapeutic agent cisplatin, which does not have any overlapping adverse effects with HCPT, was chosen to serve as the second therapeutic agent in this system (Figure 3) [34,56]. Cisplatin can chelate with carboxylic acid on the HP through non-covalent interactions and can then be released to induce the synergistic effects with HCPT [34]. According to the TEM results [34], HP can self-assemble to form short fibers with a diameter range of 6 to 9 nm and a length shorter than 1 µm. After chelation with cisplatin, the critical micelle concentration (CMC) dramatically decreased, which was closely related to the concentration of cisplatin in the mixture system. With the addition of 1.0 equiv. cisplatin, both the diameter and length of the nanofibers were enhanced in comparison to nanofibers formed by HP (Figure 3, Complex **1**), although with the addition of 1.5 equiv. cisplatin, nanoparticles with a diameter of around 23 ± 2 nm were formed (Figure 3, Complex **2**). The authors demonstrated that the nanostructures can be uptaken into the cells via caveolae-mediated endocytosis, and that cisplatin can be constantly released [34]. The IC_50_ values of HP–cisplatin Complex **1** and Complex **2** were much lower than those of cisplatin and HCPT towards a different cancer cell line, such as A549 human liver cancer cell lines, which suggests that their treatment efficacy is much better than both cisplatin and HCPT, respectively [34]. The results also showed that HCPT and cisplatin have synergistic effects against tumor cells, because both of them react with the DNA in the nucleus [34], whereby the peptide section plays a crucial role, covalently conjugating with HCPT and non-covalently interacting with cisplatin, which makes HCPT and cisplatin perform more synergistically than when simply mixing them together. The diphenylalanine part in the hexapeptide FFERGD enhances the self-assembly ability of the HP–cisplatin, while ERGD chelates with positively charged cisplatin and improves the solubility of Complex **1** and Complex **2** in physiological environments. Afterwards, Yang and co-workers conjugated HCPT with the peptide FFYpG and TSFAEYWNLLSP (PMI) to form HCPT-FFYpG-PMI [57]. In this designed molecule, HCPT is the hydrophobic head and PMI is the hydrophilic tail, which can regulate the activity of p53 via binding MDM2 and MDMX in the cell nucleus with good nuclear permeability [58]. However, the stability of the dodecapeptide (PMI) is very poor and it can be easily hydrolyzed by numerous kinases in cells [58]. The middle part of the molecule FFYpG is the trigger, while the phosphate group can dramatically enhance the solubility of this molecule. During the experiment, an interesting phenomenon was observed. Using the enzyme-instructed self-assembly strategy, leaving HCPT-FFYpG-PMI with alkaline phosphatase (ALP) in vitro under different temperatures, the resulting products had different efficiencies against cancer cells [57]. After cleaving by phosphatase at 4 °C in vitro, the balance of the hydrophobicity and hydrophilicity of this compound was broken and the resulting molecules self-assembled to form nanofibers with α-helix conformation [57]. This product has the lowest IC_50_ in this system, which is 0.22 µM towards HepG2 human liver cancer cell lines, around 10 times lower than that of HCPT [57]. Although the IC_50_ of the product produced by mixing HCPT-FFYpG-PMI with ALP at 37 °C in vitro towards HepG2 human liver cancer cell lines is higher than the product produced at 4 °C, it is still lower than that of HCPT [57]. Moreover, the nanostructures formed from this compound under different temperatures have been proven to possess very good stability and a longer half-life in situ in comparison with HCPT [57].

## 3. Conjugates of Peptides with Hydroxychloroquine (HCQ)

HCQ is a type of anti-malarial agent used for uncomplicated malaria therapy, which was approved by the FDA in 1955. The exact mechanism of HCQ is still unknown, although so far HCQ has been proven to accumulate in lysosomes and to cause cellular pH increasement, which further interferes with antigen processing in cells to reduce the inflammatory response. Additionally, HCQ has been investigated for the therapy of SARS-CoV-2 for a long time, and in 2020 it was authorized by the FDA for anti-COVID-19 therapy for a short period of time [59], since some fatalities were reported while patients were being treated with HCQ. Nowadays, different modifications of HCQ (Figure 3) for the treatment of cancer are also being extensively studied [60,61]. HCQ can connect to well-designed peptides and certain special dyes to self-assemble to form nanostructures and be used for the photothermal treatment (PTT) of cancer cells. In comparison with the conventional PTT strategy, peptide-modified materials show minimized damage toward healthy tissues caused by local hyperthermia (about 50 °C) [62]. Recently, Liang et al., designed a peptide derivative FFKYp (Yp) that covalently connects with HCQ and cypate (Cyp) via a succinate acid linker and an amide bond, respectively [62]. Cyp-HCQ-Yp can be used for the PTT of some cancers with overexpression of both ALP and carboxylesterase (CES) (Figure 4). Phosphorylated tyrosine (Yp) has the crucial function of improving the solubility of the whole compound. Additionally, phosphate on this compound can be cleaved by ALP to generate Cyp-HCQ-Y and to promote self-assembly on cancer cells. The side chain of lysine serves as a linker to connect HCQ. FFKY itself is also a brilliant assembly unit. After the enzymatic catalyzation by ALP, the compound Cyp-HCQ-Y can self-assemble to form nanoparticles with a size range of 95.1 to 169.7 nm [62]. The formed nanoparticles can be uptaken by tumor cells through endocytosis. Mammalian CES is overexpressed in hepatoma cells [63], which can metabolize an endogenous ester bond used to link HCQ to Cyp-HCQ-Yp. After being released from the nanoparticles through enzymatic stimulation, HCQ can induce autophagy inhibition, which result in the photothermal sensitization of cancer cells. Cyptate is a NIR dye that has been applied in preclinical therapy, normally as a probe. In Liang’s work, cypate had the function of harvesting light energy and transferring it to thermal energy in order to inhibit tumor cell proliferation [62]. The researchers used the dual-enzyme-controlled self-assembly system, and HCQ was used as the factor to lower the temperature requirements for PTT so as to alleviate the side effects. Basing on the results of cell assays and animal trials, PDC-Cyp has a fascinating anti-cancer ability for HpG2 cells (human liver carcinoma cells), with almost neglectable side effects compared to Cyp-Y and HCQ [62]. PTT for anti-cancer therapy has received a lot of attention in the last ten years, and Liang was one of the first researchers to combine PTT with PDC and to obtain satisfactory results. To the best of our knowledge, some PTTs used for anti-cancer therapy are currently undergoing clinical trials.

## 4. Paclitaxel

From is one of the most popular natural antineoplastic agents, which has been used for the treatment of a plethora of cancers since it was approved by the FDA. As previously mentioned, CPT was co-discovered by Professor Monroe Eliot Wal and Professor Mansukh C. Wani, and from was also co-discovered by them [35]. In 1971, from was extracted from the Pacific yew tree, and it is still one of the hardest drugs to chemically synthesize around the world at the industrial level. From restricts the function of microtubule growth by combining the β subunit of tubulin, which is the building block of the microtubule [64]. After binding with paclitaxel, tubulins are hyper-stabilized and cannot be disassembled and interfere with cell mitosis [64]. According to the clinical trials and the feedback given by the treated patients, low dosages using paclitaxel demonstrated that it would not cause severe side effects in patients [65]. Since paclitaxel already has great anti-cancer ability, there has been little research on how to modify paclitaxel to enhance its anti-tumor ability over the last five years (Figure 4). In 2009, Xu et al., first introduced the concept of using hydrogels formed by the conjugation of a peptide with paclitaxel for cancer therapy [66]. Paclitaxel was covalently linked to the peptide derivative Nap-FFKYp (1-P) on the side chain of lysine via an SA linker [66]. After the phosphate group on 1-Paclitaxel-P was cleaved by ALP, the compound Nap-FFKY-paclitaxel (1-Paclitaxel) was able to self-assemble to form nanofibers and turned into translucent hydrogel in water [66]. The results of the cytotoxicity assay demonstrated that the IC_50_ of 1-Paclitaxel-P is similar to that of paclitaxel itself, a concentration that is lower than the CMC of 1-Paclitaxel. However, in real anti-cancer therapy, the model is very different from that of the cell experiments. After conjugating paclitaxel with the peptide derivate, according to the EISA, the site-specific targeting ability of 1-Paclitaxel-P will be much better than that of paclitaxel [66]. Additionally, after the nanostructure has been built, paclitaxel can be slowly released from the system, which means that its effective time will be longer than that of the same administration dose of paclitaxel. As previously mentioned, HepG2 cells can overexpress CES, which can break the SA linker between paclitaxel and the peptide derivative to release paclitaxel [62]. Hence, 1-Paclitaxel-P may possess much better efficacy than paclitaxel itself for human liver cancer therapy.

## 5. Doxorubicin

Doxorubicin (DOX) belongs to the anthracycline class of chemotherapeutic agents, which can be isolated from antibiotics and natural sources. It can be used to treat various diseases such as acute lymphoblastic leukemia, Hodgkin’s disease, and soft tissue and bone sarcoma [67]. DOX can be employed in a plethora of strategies to inhibit tumors. It mostly interferes with the normal DNA activity by stabilizing DNA–topoisomerase II complexes or forming complexes with DNA via intercalation between DNA base pairs [67]. Nevertheless, DOX has bad site-selectivity towards cancer cells and can cause side effects to different degrees [67]. Figure 5 illustrates DOX and its derivatives that are mentioned in this part. In 2018, Chen et al., designed and synthesized the compound DOX-KGFRWR to synergistically attack cancer cells (Figure 5) [68]. In order to from this compound, DOX conjugates with the hexapeptide KGFRWR via an SA linker. KGFRWR is an amphipathic hexapeptide, which similarly to KLVFF, was obtained from β-amyloid and can self-assemble into nanofibers [50]. After conjugation with the hydrophobic drug DOX, the DOX-KGFRWR can self-assemble into nanofibers with a lower CMC and wider average diameter than nanofibers built from KGFRWR [68]. The nanofibers built from DOX-KGFRWR can allow sustained release of DOX in tumor microenvironments to increase its half-life in the presence of CES, while slowly disassembling to release a single molecule of DOX-KGFRWR for tumor inhibition. According to the results of the MTT assay, KGFRWR has an almost neglectable toxicity for SMMC-7221 cells (derived from an Asian male patient with hepatocellular carcinoma) in comparison to DOX and DOX-KGFRWR [68]. However, from the results of the MTT assay, DOX presents better anti-tumor cell ability than DOX-KGFRWR, although the results from animal trials showed the opposite outcome [68]. Before being conjugated with each other, neither DOX nor KGFRWR can inhibit the metastasis of cancer cells. Surprisingly, DOX-KGFRWR displays the ability to inhibit the activity of the extracellular enzyme metalloproteinase-2 (MMP-2), with an IC_50_ towards MMP-2 of 5.27 µM [68]. However, the mechanism of DOX-KGFRWR’s reaction with MMP-2 is still unclear. The MMP-2 concentration is high in the tumor microenvironment and plays a crucial role in inhibiting cancer metastasis [68]. Thus, after disassembly, DOX-KGFRWR has synergistic effects in anti-tumor therapy; it binds with the extracellular enzyme MMP-2 to inhibit the migration of cancer cells, and it also crosses the cell membrane via endocytosis and enters the nucleus to downregulate cancer cell mitosis.

In 2020, Guan et al., covalently combined the proapoptotic peptide (KLAK) with DOX via an acid-sensitive linker hydrazone to form KLAK-DOX [69]. Since the microenvironment of tumor tissues is acidic, KLAK-DOX can enhance the delivery efficiency of the chemotherapeutic agent. Additionally, KLAK can be released from the system to inhibit cancer cells via its own mechanism in order to achieve synergistic effects together with DOX. However, whether KLAK-DOX can self-assemble into nanostructures or not is not mentioned in their work. On another note, Ulijn et al., used a non-covalent strategy for combining DOX with a MMP-9-responsive peptide for anti-cancer therapy [70]. The anti-cancer efficacy of the non-covalently modified DOX was also enhanced based on the results of the animal trials.

## 6. Dexamethasone (Dex)

As previously mentioned, Liang and co-workers designed the molecule Cyp-HCQ-Yp through a PDC strategy for anti-cancer therapy [62]. However, they initially used this strategy to modify dexamethasone to increase its anti-hepatic fibrosis effect in 2018 [71]. Dexamethasone is an FDA-approved drug used for treating many inflammatory conditions, such as bronchial asthma, endocrine inflammation, and gastrointestinal inflammation [72] Additionally, Dex was approved for the treatment of severe respiratory symptoms caused by COVID-19 in 2020 [73]. Figure 6 summarized Dex and its derivatives that are mentioned in this part. In the following case, the peptide derivative Nap-FFKYp (1-P), which was introduced in Section 4, was used again to covalently interact with Dex through an SA linker on the side chain of K to 1-Dex-P [71]. Figure 6 shows the speculated mechanism of 1-Dex-P against anti-hepatic fibrosis. With the phosphate group, 1-Dex-P has very good water solubility. Moreover, ALP is an important marker for liver diseases, because its concentration is higher in liver fibrosis microenvironments than in normal tissues [71]. After the cleavage of the phosphate group on 1-Dex-P by ALP in the special microenvironment, its (1-Dex) hydrophilicity sharply declined and it was able to self-assemble into nanofibers to form a hydrogel through the EISA strategy [71]. Nonetheless, Dex was not entrapped on the nanofibers. The speculated location of Dex is on the surface of the nanofibers in vitro. Due to the overexpression of esterase during hepatic fibrosis inflammation, Dex can be released by the esterase from the nanofibers via hydrolysis [71]. Although Dex is released, the leftover parts formed by Nap-FFKY are still nanofibers but with shorter diameters [71]. After being released, Dex can inhibit the activation of hepatic stellate cells (HSC), which play a crucial role in inducing hepatic fibrosis. The authors also found that if the administration dosage of 1-Dex-P is lower than the CMC value of 1-Dex, its anti-hepatic fibrosis efficacy is not enhanced in comparison with treating it with Dex alone [71], suggesting the importance of the self-assembly of the prodrug. 1-Dex-P has been proven to have low toxicity to normal tissues, according to the results of an H&E staining assay [71]. Moreover, liposomes were also utilized to non-covalently interact with and deliver Dex in order to improve Dex’s working efficiency.

## 7. Conclusions and Perspectives

Assemblies of PDCs have been widely studied to enhance the efficacy and efficiency of therapeutic agents, especially in cancer therapy, and much progress has been made in recent years. Normally, the self-assembly of PDCs improves the efficiency of the original drugs through the following strategies. First, PDCs have the ability to self-assemble or undergo an instructed self-assembly through molecular engineering. This characteristic can help PDCs form nanostructures to slowly release drugs under specific stimuli in the microenvironment of the diseases (e.g., enzymatic reactions or redox reactions). Secondly, the nanostructures formed by PDCs can be used as nanocarriers to delivery other drugs, proteins, or even mRNAs to induce synergistic effects for various treatments. Thirdly, an instructed self-assembly strategy (e.g., EISA) can assist PDCs to accumulate in the specific target of treatment to improve the selective targeting of drugs. Moreover, PDC strategies can reduce the side effects induced by the drugs because of the good biocompatibility and adaptive biodegradability of PDCs with lower administration dosages.

Although the development of the self-assembly of PDCs has rapidly progressed in recent years, no product has yet been approved by the FDA, not even for clinical trials. There are still many urgent issues that need to be addressed. The mechanisms of most drugs used for PDCs are clear, although the working mechanisms of the peptides are still unclear or unknown, even though peptides are biocompatible. The intracellular environments and microenvironments of different tissues are complicated, which makes the pathways for PDCs harder to reveal. Additionally, costs need to be carefully considered if self-assembling PDCs are to be manufactured, taking into account that the costs for PDCs are higher than those of other small molecular drugs and that their clinical trials are very expensive.

Since human insulin was approved by the FDA in 1982, more protein drugs have been approved by the FDA in the 21st century. Therefore, we believe that PDCs have a bright future in clinical usage, because most sequences of peptides in PDCs come from proteins or slight modifications in the peptide sequence. Overall, PDCs are an interesting area of study, which still has many treasures waiting to be discovered by scientists from different disciplines and which will keep bringing us surprises.

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
