# Peer review of "Molecular Engineering of Peptide–Drug Conjugates for Therapeutics"

_pharmaceutics, 2022, doi:10.3390/pharmaceutics14010212_

Round 1

Reviewer 1 Report

The manuscript “Molecular Engineering of Peptide-Drug Conjugates for Therapeutics” investigates on self-assembled nanostructures of PDCs for cancer therapy. The authors focus on the recent development of self-assembled nanostructures of PDCs. The manuscript is interesting in the context of cancer therapy. However authors should explain and investigate better on the mechanisms how drugs and peptides, which are arranged together to form diverse nanostructures, may enhance both efficiencies and efficacies of those drugs  Further information from in vitro and in vivo studies should be provide by the authors in order to understand the potentialities of these complex compounds in cancer therapy.  A more accurate description on the side effects of these compounds, and of the molecular mechanisms mediate their effects will be helpful for a better understanding of how they can have a bright future in clinic.

Major concerns.

For instance, the authors should explain better how CDA and CPT released from the hydrogel reservoir can activate both the adaptive and the innate immune system. What is the release efficiency of CDA? Could the authors give more information on its cytotoxic effects on cancer cells in this experimental setting?

The authors claim that both CDA-NT and P-NT can induce a memory response. The authors should explain better this point. What is the toxicity of these compound on heathy tissues?

HCPT causes patient’s compliance. What are the side-effects of complex 1 and complex2? The authors should better investigate the experimental models that have been used to test these compounds.

Can the author give more accurate information about the HCPT-FFYpG-PMI? What are the effects on healthy tissue? Which phosphatase is involved in the cleavage? What is the cytotoxic effect on cancer cells?

The authors claim that the target specificity of 1-Paclitaxel-P was greatly enhanced in comparison to Paclitaxel. However, both compounds show similar IC50. What is the advantage of 1-Paclitaxel-P compared to Paclitaxel? Are there in vivo data that might shed light on 1-Paclitaxel-P?

DOX presents better anti-tumor cell ability than DOX-KGFRWR. However, DOX-KGFRWR, by inhibiting MMP-2, inhibits metastasis, but both Dox or KGFRWR are not able to inhibits MMP-2. What is the molecular mechanism of inhibition of DOX-KGFRWR on MMP-2? Is DOX-KGFRWR able to bind MMP-2? The authors should give further information on the cytotoxicity DOX-KGFRWR in comparison to DOX.

Reviewer 2 Report

The manuscript summarizes some recent findings regarding the encapsulation and use of chemotherapeutic drugs in peptides and nanostrutucred complexes. On overall, I found the review reasonably organized, albeit not disrupting in terms of reasoned summary of the literature. The authors should improve the english which is rather poor, but otherwise the review is worth publication.

Reviewer 3 Report

This manuscript reviews the current literature on the development and molecular engineering of Peptide-drug conjugates for therapeutic applications. This is an interesting topic, and as the authors mention, there are so many important factors to consider in characterizing the properties of these systems.

My main concerns are the following:

  • The English quality is acceptable, but I would suggest to have the paper proof (re-)read by a native speaker. Some of the sentences are unclear and in general, the construction of sentences could be improved. Moreover, the text contains many typos (ex: line 12: “high undeirable toxicity”; line 12: “Also, many cancer therapeutic agents are small molecular drugs, while a variety of cancer cells have drug resistence or multiple drug resistance (MDR) after long-term therapy”: does not make sense at all; line 15: “precise locolization”. Many more throughout the paper). The organization is not clear and somehow hard to follow
  • The review mostly summarizes source articles, without providing a critical analysis that makes sense of the collection of the cited articles. A good literature review should analyze, synthesize, and critically evaluate the cited papers to give a clear picture of the state of knowledge on the subject
  • Although the authors seem to have permission from the concerned editors, the figures are mere reproductions of previously published figures and therefore lack of homogeneity between them

In conclusion, while the decision to publish this review should be left to the editor, I think it contains too many drawbacks to be published in Pharmaceutics, at least on its current form.

Round 2

Reviewer 1 Report

This second version of the manuscript has improved. The authors have answered all the suggestions of the reviewer. From my point of view the manuscript is suitable for publication.

Reviewer 3 Report

The authors have substantially improved the core of the manuscript. I still think they could have modified the figures in a way that could have made them more homogenous between each other, as there are still mere reproductions of the source articles. However, if it’s not an issue for the Editor, the paper can be published in Pharmaceuticals.